# Megafaunal extinctions, not climate change, may explain Holocene genetic diversity declines in *Numenius* shorebirds

Hui Zhen Tan[1], Justin JFJ Jansen[2], Gary A Allport[3], Kritika M Garg[1†, ‡], Balaji Chattopadhyay[1§], Martin Irestedt[4], Sean EH Pang[1], Glen Chilton[5], Chyi Yin Gwee[1#], Frank E Rheindt[1*]

[1]Department of Biological Sciences, National University of Singapore, Singapore, Singapore; [2]Naturalis Biodiversity Center, Leiden, Netherlands; [3]BirdLife international, Cambridge, United Kingdom; [4]Department of Bioinformatics and Genetics, Swedish Museum of Natural History, Stockholm, Sweden; [5]Department of Biology, St. Mary's University, Calgary, Canada

**\*For correspondence:**
dbsrfe@nus.edu.sg

**Present address:** [†]Centre for Interdisciplinary Archaeological Research, Ashoka University, Sonipat, India; [‡]Department of Biology, Ashoka University, Sonipat, India; [§]Trivedi School of Biosciences, Ashoka University, Sonipat, India; [#]Division of Evolutionary Biology, Faculty of Biology, Ludwig Maximilian University of Munich, Planegg-Martinsried, Germany

**Competing interest:** The authors declare that no competing interests exist.

**Abstract** Understanding the relative contributions of historical and anthropogenic factors to declines in genetic diversity is important for informing conservation action. Using genome-wide DNA of fresh and historic specimens, including that of two species widely thought to be extinct, we investigated fluctuations in genetic diversity and present the first complete phylogenomic tree for all nine species of the threatened shorebird genus *Numenius*, known as whimbrels and curlews. Most species faced sharp declines in effective population size, a proxy for genetic diversity, soon after the Last Glacial Maximum (around 20,000 years ago). These declines occurred prior to the Anthropocene and in spite of an increase in the breeding area predicted by environmental niche modeling, suggesting that they were not caused by climatic or recent anthropogenic factors. Crucially, these genetic diversity declines coincide with mass extinctions of mammalian megafauna in the Northern Hemisphere. Among other factors, the demise of ecosystem-engineering megafauna which maintained open habitats may have been detrimental for grassland and tundra-breeding *Numenius* shorebirds. Our work suggests that the impact of historical factors such as megafaunal extinction may have had wider repercussions on present-day population dynamics of open habitat biota than previously appreciated.

## Editor's evaluation

This study uses genomic inferences to reconstruct past population sizes of whimbrel and curlew shorebirds, along with niche modeling approaches, to explore changes in those populations over millenia. Steppe-dependent breeding species appear to have declined more prominently than species that breed in other habitats. The coincident timing of these declines of steppe-dependent breeding shorebirds, and the extinction of the mammalian megafauna that likely maintained that habitat, raises the intriguing possibility that those mammalian extinctions had broad effects on these shorebirds and the entire community of steppe-dependent organisms.

## Introduction

Rates of population decline and extinction have risen sharply during the ongoing sixth mass extinction crisis (*Ceballos et al., 2020*; *Dirzo and Raven, 2003*; *Sánchez-Bayo and Wyckhuys, 2019*; *Stuart et al., 2004*). Species distribution models based on future climate scenarios forecast that rates of

**eLife digest** About 20,000 years ago, the Earth was a much colder world roamed by giant mastodons, gigantic elks, woolly mammoths and sabre-tooth tigers. Yet these imposing creatures were living on borrowed time: by the start of the Holocene, around 10,000 years later, many animals over 45kg had vanished across the Northern Hemisphere, closing the book on what is known as the Quaternary extinction event. As large grazers disappeared, the landscape likely changed too. Where open tundra and grasslands may have once dominated, woodlands and shrubs probably took over, creating ripple effects for surviving species.

These extinction events took place in a changing world, with glaciers starting to retreat about 20,000 years ago and human populations colonizing an increasing share of this planet's land area. In fact, since the end of this last glacial maximum, ecosystems have been reshaped by a succession and a combination of climatic, historical and human-driven forces. This makes it difficult for scientists to disentangle the relative contribution of these factors on the lives of animals.

Tan et al. decided to explore this question by reconstructing how effective population sizes changed over the past 20,000 years for nine species of curlews and whimbrels. These shorebirds, which together comprise the genus *Numenius*, breed slowly and nest in open environments such as moorlands or tundra. Many are currently under threat.

Fluctuations in the numbers of breeding individuals affect the genetic diversity of a species, and these events leave tell-tale genetic signatures that can be uncovered through DNA analyses. Tan et al. had enough fresh and museum samples to infer these changes for five *Numenius* species, revealing that genetic diversity brutally dropped soon after the last glacial period ended.

At the time, humans were yet to make significant changes on their environment and a warming world should have supported population growth. Tan et al. suggest that, instead, this sharp decline is linked to the late Quaternary extinctions of large mammals: with the demise of grazing animals which could keep woodlands at bay, the shorebirds lost their open nesting grounds. This event has left its mark in the genome of existing species, with these birds still exhibiting a low level of genetic diversity that may put them at further risk for extinction.

endangerment will further accelerate, underscoring the need for conservation action (*Thomas et al., 2004*). In this era of increasing biodiversity loss, the maintenance of genetic diversity within species has become a focus of conservation as it is thought to predict evolutionary adaptability and extinction risk (*Frankham, 2005*; *Hoban et al., 2020*; *Jetz et al., 2014*). Modern declines in genetic diversity have been documented for a handful of species (*Allentoft and O'Brien, 2010*; *Chattopadhyay et al., 2019*; *Evans and Sheldon, 2008*; *Garner et al., 2005*), but we continue to know little about the global mechanisms of genetic diversity loss.

Anthropogenic climate change is widely recognized for its pervasive impact on biodiversity and genetic diversity (*Johnson et al., 2017*; *Miraldo et al., 2016*; *Turvey and Crees, 2019*). However, historical events have equally left their signature in the genetic profiles of present-day species (*Hewitt, 2000*). Comparative genomics of extinct versus extant species could add an important perspective to elucidating such trends in faunal endangerment (*Frankham, 2005*).

We used a museomic approach to investigate fluctuations in effective population size in all nine species of the migratory shorebird genus *Numenius*, known as whimbrels and curlews, including two species, the slender-billed curlew (*N. tenuirostris*) and Eskimo curlew (*N. borealis*), that are presumed to be extinct (*Buchanan et al., 2018*; *Butchart et al., 2018*; *Kirwan et al., 2015*; *Pearce-higgins et al., 2017*; *Roberts et al., 2010*; *Roberts and Jarić, 2016*). Members of the genus *Numenius* breed across the Northern Hemisphere's tundras and temperate grasslands, and are particularly vulnerable to endangerment due to comparatively long generation times (*Pearce-higgins et al., 2017*).

Our objective was to characterize genetic diversity fluctuations in *Numenius* shorebirds, assess the relative impact of historical and anthropogenic factors on these fluctuations, and determine the mechanisms that may have had the biggest impact on their populations. Because of their dependence on open habitats, we expected the genetic diversity trends of whimbrels and curlews to track the availability of such habitats across the Late Quaternary. We also expected significant declines in genetic diversity during the late Holocene when global human activity intensified, not least because

the demise of the two extinct species has been attributed to habitat loss and hunting (**Committee on the Status of Endangered Wildlife in Canada, 2009**; **Gallo-Orsi and Boere, 2001**). By testing the timing of genetic diversity fluctuations against that of important ecological events, we homed in on the factors that influenced the evolutionary trajectory of this threatened shorebird lineage over the last ~20,000 years.

## Results and discussion

We sequenced 67 ancient and fresh samples across all nine *Numenius* species for target enrichment (**Figure 1A**; **Supplementary file 1**). After filtering for quality, a final alignment of 514,771 bp across 524 sequence loci was retained for each of the 62 remaining samples at a mean coverage of 118 X. Phylogenomic analyses using MP-EST (**Liu et al., 2010**) revealed two separate groups with high support, here called the 'whimbrel clade' and the 'curlew clade,' that diverged approximately 5 million years ago (**Figure 1B**; **Figure 1—figure supplement 1A**). This is the first phylogenomic tree to include all members of the genus *Numenius*. The use of degraded DNA from toepads of museum specimens allowed us to include the two presumably extinct taxa. Of these, the slender-billed curlew emerged as sister to the Eurasian curlew (*N. arquata*), a phenotypically similar species that occurs in sympatry in Central Asia (**Sharko et al., 2019**). On the other hand, the Eskimo curlew emerged as a distinct member of the curlew clade with no close relatives (**Figure 1B**). Our phylogenomic dating analyses demonstrated that 40.6% of the evolutionary distinctness (**Jetz et al., 2014**) of the curlew clade has been lost with the presumable extinction of the two species, and that another 15% is endangered (**Figure 1B**; **Supplementary file 2**).

To characterize the differential impacts of extinction pressures, we reconstructed the demographic history of *Numenius* shorebirds. For five species with a sufficiently high sample size, we employed stairway plots (**Liu and Fu, 2020**) to infer fluctuations in effective population size ($N_e$), a proxy for genetic diversity, given that this method works well for reduced representation genomic datasets such as ours, and has a relatively high accuracy for reconstructions of diversity change in the Late Quaternary (**Liu and Fu, 2020**). Fluctuations in $N_e$ were compared against key biotic and anthropogenic events of the Late Quaternary. We also accounted for climatic changes by modeling the extent of suitable breeding areas of each species under climate conditions prevalent during the present-day (1960–1990), mid-Holocene (6,000 years ago), and Last Glacial Maximum (LGM; 22,000 years ago) using the Maxent algorithm (**Phillips et al., 2006**).

The Last Glacial Period preceding the LGM saw ice sheets at their maximum extent (**Hughes et al., 2013**). During this time, tundra habitats dominated the northern latitudes and an increase in $N_e$ in the tundra-inhabiting Eurasian whimbrel (*N. phaeopus*) was observed (**Binney et al., 2017**; **Wang et al., 2021**; **Zimov et al., 1995**; **Figure 1C**). Soon after, during the Pleistocene-Holocene transition, our stairway plots revealed generally sharp declines of $N_e$ in most species despite an increase in the area of suitable breeding habitat predicted (**Figure 1C**). The extent of breeding habitat predicted by our ecological niche models relied on bioclimatic variables, suggesting that – paradoxically – favorable conditions for *Numenius* shorebirds in the lead-up to the Holocene did not trigger an increase in genetic diversity, but instead coincided with precipitous declines in $N_e$. A decrease in $N_e$ could be expected during the period when most species underwent rapid range expansion shortly after the LGM (**Braasch et al., 2019**). However, $N_e$ declines in all species persisted beyond the mid-Holocene up until a period when habitat availability started to resemble the levels that were prevalent just before the Anthropocene (**Figure 1C**; **Figure 1—figure supplement 3**). Therefore, the Holocene collapse of genetic diversity in *Numenius* shorebirds cannot be explained purely by range expansions. To understand the drivers of $N_e$ declines in *Numenius* shorebirds, factors other than climate change would need to be considered.

During the Pleistocene-Holocene transition (starting at roughly 20,000 years ago), a mass extinction of megafaunal mammals (≥44 kg) was underway, known as the Late Quaternary Extinctions (**Hedberg et al., 2022**; **Johnson, 2009**), with most becoming extinct by 10kya (**Figure 1C**; **Koch and Barnosky, 2006**; **Stuart, 2015**). Megafaunal mammals are ecosystem-engineers that maintain open landscapes such as temperate grasslands and steppes through grazing, browsing, and physical impacts (**Bakker et al., 2016**; **Goheen et al., 2018**). During the intervening period between their extinction and the spread of ungulate domestication, there would have been no functional replacements for these ecosystem services (**Hedberg et al., 2022**; **Lundgren et al., 2020**). Open grasslands

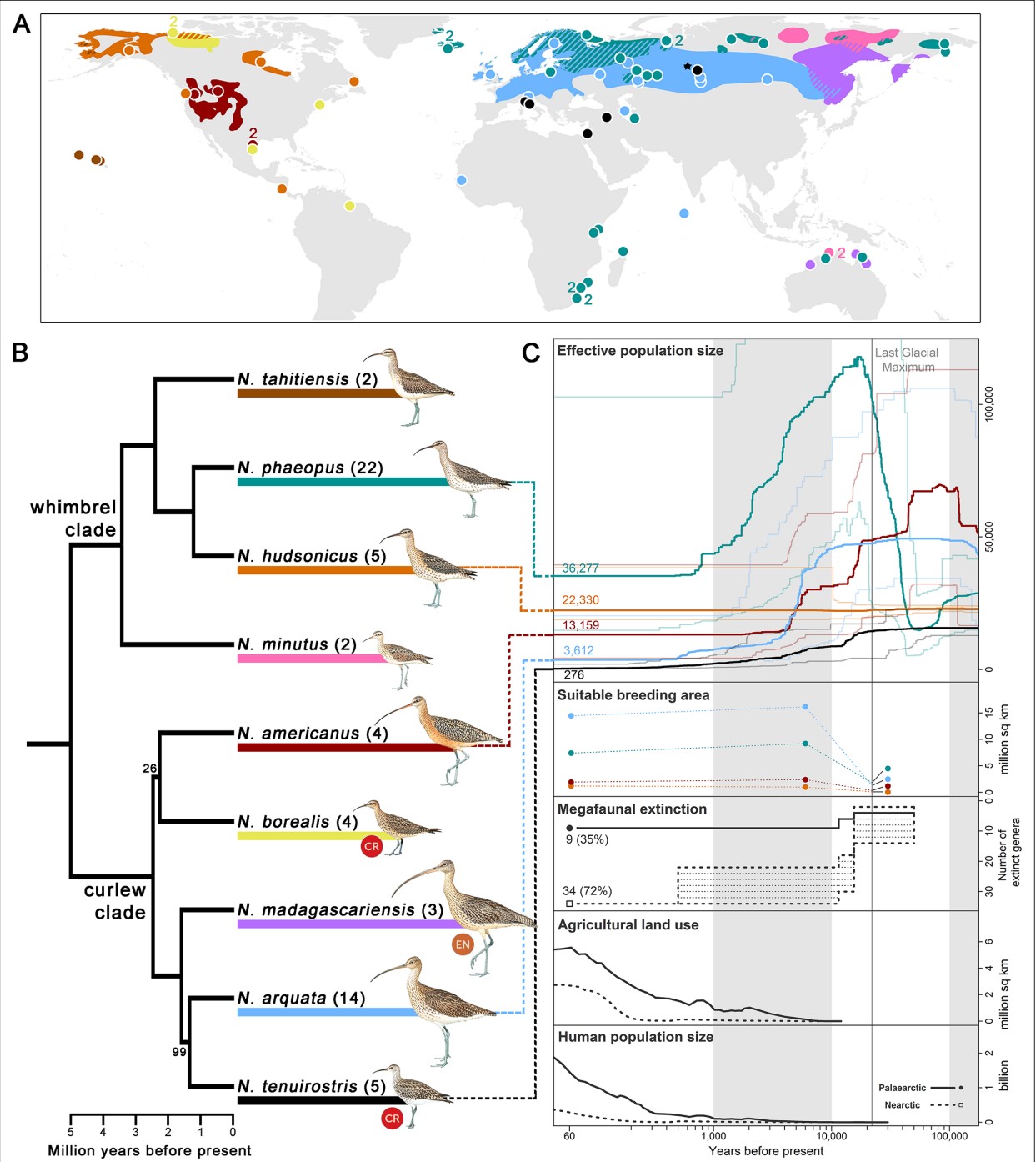

**Figure 1.** *Numenius* phylogenomic relationships and Quaternary population trajectories. (**A**) Breeding distribution map and sampling localities of each *Numenius* species (***BirdLife International and Handbook of the Birds of the World, 2017***; ***Lappo et al., 2012***); wintering and migratory ranges are not shown. Colors correspond to species identities in (**B**). Diagonal lines denote regions with co-distributed species. Each circle represents one sample unless otherwise specified by an adjacent number. The only known breeding records of *N. tenuirostris* were from near the village of Krasnoperova c.10 km south of Tara, Omsk (Russia), which is denoted by a black star (★), although this might not have been the core breeding area. (**B**) Phylogenomic tree constructed from an alignment of 514,771 bp across 524 sequence loci. Tree topology (including bootstrap support values) and divergence times were estimated with MP-EST and MCMCTree, respectively. Only bootstrap <100 is displayed. Sample sizes for each species are given in brackets. IUCN Red List status of critically endangered (CR) and endangered (EN) species is indicated. (**C**) Results of demographic history reconstruction using stairway plot for selected species displayed with key climatic, biotic, and anthropogenic events. Effective population size: Line colors correspond to species identities in the tree in (**B**) and numbers at present time represent present-day effective population sizes. Thick lines represent the median effective population size while thin lines represent the 2.5 and 97.5 percentile estimations. The vertical gray line denotes the Last Glacial Maximum (LGM) and panels are

*Figure 1 continued on next page*

*Figure 1 continued*

shaded to aid reference to the time axis. Suitable breeding area: predicted suitable breeding area at LGM (22,000 years ago), mid-Holocene (6,000 years ago), and present-day (1960–1990) using Maxent. Dot colors correspond to species identities in the tree in (**B**). Dotted lines connecting the dots are for visualization purposes and do not represent fluctuations in the breeding area. The following panels display the timings of key climatic, biotic and anthropogenic events, including megafaunal extinction (in terms of the number of extinct genera with dotted shading denoting uncertainty in estimates; *Koch and Barnosky, 2006*), agricultural land use, and human population size (HYDE 3.2; *Klein Goldewijk et al., 2017*; *Klein Goldewijk et al., 2010*). Line type corresponds to geographical area (Nearctic versus Palaearctic) as denoted in the 'Human population size' panel.

The online version of this article includes the following figure supplement(s) for figure 1:

**Figure supplement 1.** Principal component (PC) analysis of *Numenius* samples, with the percentage of variation of the two most important PCs displayed.

**Figure supplement 2.** Demographic history reconstruction using stairway plot for *N. borealis*, showing results for two datasets, one containing all five samples and the other a subset of three samples with low missingness.

**Figure supplement 3.** Visualization of the ecological niche model results in green corresponding to a higher probability of presence and brown corresponding to a lower probability.

would have been subject to increasing forest succession (*Johnson, 2009*) and the amount of suitable habitat for *Numenius* shorebirds might have been less than predicted by forecasts relying only on bioclimatic variables. Therefore, genetic diversity fluctuations in *Numenius* shorebirds run counter to expectations based on natural climate change and seem to be better explained by the demise of the ecosystem-engineers that would have helped maintain shorebird breeding habitats.

By the late Holocene, the genetic diversity of most *Numenius* shorebirds stabilized at a time when anthropogenic impact was only starting to expand across the Northern Hemisphere with a steep rise in human population and land conversion for agriculture (*Figure 1C*). The timing of these events is inconsistent with the hypothesis that direct anthropogenic activity has been the main cause of genetic diversity declines in *Numenius* (*Crisp et al., 2011*). Events unrelated to modern anthropogenic pressure seem to have played a bigger role in the diversity declines observed in *Numenius* shorebirds (*Lucena-Perez et al., 2020*; *Nadachowska-Brzyska et al., 2015*). It is possible that additional adverse effects caused by more recent anthropogenic impacts are not yet reflected in the genomes investigated, perhaps exacerbated by shorebirds' long generation times.

At present, members of the curlew clade, which predominantly breed in temperate grasslands at lower latitudes, generally exhibit levels of $N_e$ that are lower than those of the higher latitude-breeding whimbrels (*Figure 1C*). Temperate grasslands face far greater anthropogenic pressures from land use than the northerly tundra (*Pimm et al., 2014*), contributing to further declines in curlews more so than in whimbrels. Strong differences in the demographic histories uncovered within the whimbrel clade (specifically between *N. phaeopus* and *N. hudsonicus*) probably reflect the uneven distribution of glacial extent and impact across the northern hemisphere, with North America being covered by extensive ice sheets during the LGM while most of Siberia remained ice-free, allowing for a disproportionate increase of $N_e$ in *N. phaeopus*. Genetic diversity estimates were lowest in the presumably extinct slender-billed curlew *N. tenuirostris* (*Figure 1C*). Low genetic diversity may contribute to a species' extinction risk (*Frankham, 2005*; *Spielman et al., 2004*), although such links must be examined for each species independently and could possibly be conflated with other factors such as total population size (*Evans and Sheldon, 2008*; *Teixeira and Huber, 2021*).

Our study uncovers substantial declines in genetic diversity in curlews and whimbrels across the Late Quaternary. Analysing $N_e$ fluctuations over time allowed us to test which factors may have coincided with genetic diversity declines. Of the factors investigated, megafaunal extinctions—not natural climate change in the post-glacial period—best explain these declines and may have had cascading effects on species' evolutionary trajectories that continue to impact them to the present-day. Future work should examine additional factors such as non-breeding habitat availability, although this factor is unlikely to account for post-LGM diversity declines in Old World shorebirds as the total length of coastlines would have increased in areas such as Southeast Asia where rising sea levels have led to the inundation of large shelf areas and created complex archipelagos such as Indonesia (*De Groeve et al., 2022*; *Sarr et al., 2019*). Our results underscore that grassland biomes and their biota face unique challenges that warrant more conservation attention (*Ceballos et al., 2010*; *Chan et al., 2005*; *Helm*

*et al., 2009*; *Nakahama et al., 2018*; *Török et al., 2016*; *Wesche et al., 2016*). Our work demonstrates that relatively brief evolutionary events, such as the Late Quaternary Extinctions of megafauna, may have long-lasting evolutionary effects on populations, in our case for roughly ~10,000 years. The plight of *Numenius* shorebirds is a sobering reminder of the importance of conserving remaining genetic diversity to ensure the resilience of our planet's biota.

## Materials and methods

### Taxon sampling

We acquired samples for all nine species in the genus *Numenius*, encompassing most of the known subspecies. Species and subspecies identities are as provided by the source museum or institution (*Supplementary file 1*) or assigned in reference to known breeding and wintering locations (*Birds of the World, 2022*). We also included one common redshank *Tringa totanus* as an outgroup for phylogenetic rooting. All samples were acquired through museum loans except for an individual of the endangered subspecies *N. phaeopus alboaxillaris* that was sampled during fieldwork by GAA (*Supplementary file 1*). Where possible, we acquired fresh samples (tissue or blood) because of their higher DNA quality. To represent rarely-sampled or presently-rare taxa for which no fresh samples were available, we acquired toepad material from historic museum specimens and applied ancient DNA methods.

### Baits design for target capture

We used the *Calidris pugnax* genome (accession no. GCA_001458055.1) (*Küpper et al., 2015*) to design baits to capture selected exons. We used EvolMarkers (*Li et al., 2012*) to identify single-copy exons conserved between *C. pugnax, Taeniopygia guttata* (accession no. GCF_003957565.1; released by the Vertebrate Genomes Project) and *Ficedula albicollis* (accession no. GCA_000247815.1). Exons longer than 500 bp with a minimum identity of 55% and an e-value $10e^{-15}$ were isolated with bedtools 2.28.0 (*Quinlan and Hall, 2010*), forming our target loci. Only target loci with 40–60% GC content were retained and any overlapping loci were merged (*Quinlan and Hall, 2010*). Target loci with repeat elements were then filtered out in RepeatMasker 4.0.6 (*Smit et al., 2015*). We arrived at a final set of 565 unique target loci with a mean length of 970 bp. These target loci were used to design 19,003 100 bp-long biotinylated RNA baits at 4 X tiling density (MYcoarray/Arbor Biosciences, USA).

### Laboratory methods

Both fresh and historic samples were subjected to DNA extraction, followed by library preparation and target enrichment, with slight modifications for various sample types to optimize yield. DNA extractions of fresh samples were performed using the DNEasy Blood & Tissue Kit (Qiagen, Germany) with an additional incubation step with heat-treated RNase. Extractions for historic samples were performed using the same kit but with modifications (*Chattopadhyay et al., 2019*). Historic samples were washed with nuclease-free molecular grade water before extraction and dithiothreitol was added to the digestion mix. DNA precipitation was performed for at least 12 hr and MinElute Spin Columns were used for elution (Qiagen, Germany). Historic samples were processed in a dedicated facility for highly degraded specimens.

DNA extracted from fresh samples was sheared via sonification using Bioruptor Pico (Diagenode, Belgium) to a target size of 250 bp. DNA extracts from historic samples were generally smaller than the target size; hence no further shearing was performed. Whole-genome libraries were prepared using the NEBNext Ultra II DNA Library Prep Kit for Illumina (New England Biolabs, Ipswich, USA) with modifications for subsequent target enrichment. For fresh samples, adaptor concentrations were kept constant regardless of the DNA input amount. Size selection with AMPure XP beads (Beckman Coulter, USA) was performed for 250 bp insert sizes. The reaction was split into two equal parts before polymerase chain reaction (PCR) amplification and combined afterward for subsequent steps. For historic samples, a formalin-fixed, paraffin-embedded (FFPE) DNA repair step was first performed using NEBNext FFPE DNA Repair Mix (New England BioLabs). A 10-fold dilution of adaptors was used, and no size selection was performed. For both types of samples, twelve cycles of PCR amplification were performed.

Target enrichment was carried out following the MYbaits manual (Arbor Biosciences, USA) with modifications (*Chattopadhyay et al., 2019*). We used 1.1 uL of baits per fresh sample (~5 X dilution) and 2.46 uL of baits per ancient sample (~2 X dilution). For fresh samples, hybridization of baits and target loci was performed at 65 °C for 20 hr and 15 cycles of amplification were performed. For historic samples, hybridization was performed at 60 °C for 40 hr, and 20 cycles of amplification were performed. For both fresh and historic samples, one negative control sample was added for each batch of extraction, library preparation, and target enrichment. Extracts, whole-genome libraries, final enriched libraries, and all negatives were checked for DNA concentration on a Qubit 2.0 Fluorometer using the Qubit dsDNA HS assay kit (Thermo Fisher Scientific, USA), and for fragment size on a Fragment Analyzer using the HS NGS Fragment kit (1–6000 bp) (Agilent Technologies Inc, USA). Final enriched libraries were pooled at equimolar quantities. A total of 67 enriched libraries were sequenced, with fresh and historic samples sequenced separately on two Illumina HiSeq 150 bp paired-end lanes (NovogeneAIT, Singapore).

## Reference genome assembly

We obtained a sample of *N. phaeopus* (ZMUC 112728) from the Natural History Museum of Denmark, Copenhagen, for reference genome assembly. Its genomic DNA was extracted using the KingFisher Duo Prime Magnetic Particle Processor (Thermo Fisher Scientific, USA) and the KingFisher Cell and Tissue DNA Kit (Thermo Fisher Scientific). A linked-read sequencing library was prepared using the Chromium Genome library kits (10 X Genomics) and sequenced on one Illumina Hiseq X lane at SciLifeLab Stockholm (Sweden). The de novo assembly analysis was performed using 10 X Chromium Supernova (v. 2.1.1). Reads were filtered for low quality and duplication, while assemblies were checked for accuracy and coverage and the best assembly was selected based on the highest genome coverage with the fewest errors. The final genome had a size of 1.12 Gb at a coverage of 50 X with N50=3504.2 kbp.

## Raw reads processing

Raw reads were checked for sequence quality in FastQC 0.11.8 (Babraham Bioinformatics) and trimmed to remove low-quality termini and adaptors in fastp 0.20.0 (*Chen et al., 2018*). We retained reads with a minimum length of 36 bp and set a phred quality threshold of 20. Retained reads started at the first base satisfying minimum quality criteria at the 5'-end and were truncated wherever the average quality fell below the threshold in a sliding window of 5 bp. Duplicates were removed using FastUniq 1.1 (*Xu et al., 2012*) before sequence quality, duplication rate, and adaptor content were checked again in FastQC. We employed FastQ Screen 0.14.0 (*Wingett and Andrews, 2018*) to assign the source of DNA against a list of potential contaminants. We aligned reads to our assembled *Numenius phaeopus* genome, *Homo sapiens* (accession no. GCF_000001405.39), and a concatenated database of all bacterial genomes available on GenBank (National Center for Biotechnology Information (NCBI), 1988). Only reads that mapped uniquely to the *N. phaeopus* genome were retained. Reads were sorted and re-paired using BBtools 37.96 (*Bushnell, 2014*). Downstream bioinformatic procedures were split into single nucleotide polymorphism (SNP)-based and sequence-based analyses.

## SNP calling

For SNP-based analyses, reads were aligned to the target sequences used for bait design with bwa-mem 0.7.17 (*Li, 2013*). The output alignment files were converted to bam files (view) and sorted by coordinates (sort) using SAMtools 1.9 (*Li et al., 2009*). Alignments were processed in Picard 2.20.0 (Picard tools, Broad Institute, Massachusetts, USA) to add read group information (AddOrReplaceReadGroups), and another round of duplicate identification was performed (MarkDuplicates) before alignment files were indexed (BuildBamIndex). The reference file of target sequences was indexed in SAMtools (faidx) and a sequence dictionary was created in Picard (CreateSequenceDictionary). To improve SNP calling accuracy, indel realignment was performed in GATK 3.8 (*McKenna et al., 2010*) (RealignerTargetCreator, IndelRealigner). We inspected historic DNA alignments in mapDamage 2.0.9 (*Jónsson et al., 2013*) and trimmed up to 5 bp from the 3' ends of both read to minimize frequencies of G to A misincorporation (<0.1) and soft clipping (<0.2). Finally, alignments were checked for quality and coverage in QualiMap 2.2.1 (*Okonechnikov et al., 2016*).

We first generated likelihoods for alignment files in BCFtools 1.9 (*Li, 2011*) (mpileup), skipping indels. Using the same program, we then called SNPs (call) for all *Numenius* samples using the multiallelic and rare-variant calling model. Called SNPs were filtered in VCFtools 0.1.16 (*Danecek et al., 2011*) to retain sites with quality values >30, mean depth 30–150, minor allele frequency ≥0.02, and missing data <5%, in this order. Missingness and depth of sites and individuals, respectively, were quantified for SNPs called. We removed eight individuals from downstream analyses due to a combination of high missing data (>0.4%) and low coverage (<36 X), yielding a SNP set representing 58 samples. A Perl script (rand_var_per_chr.pl) was used to call one SNP per locus to avoid calling linked SNPs (*Caballero, 2018*). SNPs were further screened for linkage disequilibrium in PLINK 1.9 (*Purcell et al., 2007*) using a sliding window of 50 SNPs with a step size of 10 and an $r^2$ correlation threshold of 0.9. We also screened for the neutrality of SNPs in BayeScan 2.1 (*Foll and Gaggiotti, 2008*) using default settings. We additionally created a dedicated SNP set per species for input into demographic history reconstruction using the method described above, but without minor allele frequency cut-offs and with all SNPs at each locus retained.

## Population genomic analyses

We conducted principal component analysis (PCA) for all *Numenius* samples using the R package SNPRelate 1.16.0 (*R Development Core Team, 2022*; *Zheng et al., 2012*; *Figure 1—figure supplement 1A*). We did not detect any considerable genomic differentiation along subspecific delimitations within *N. phaeopus* and *N. arquata*, whose population-genetic structure had been resolved with thousands of genome-wide markers in a previous study (*Tan et al., 2019*; *Figure 1—figure supplement 1B, C*). Samples of *N. p. alboaxillaris* and *N. a. suschkini,* two Central Asian taxa that are described in the literature as phenotypically differentiated (*Allport, 2017*; *Engelmoer and Roselaar, 1998a*; *Engelmoer and Roselaar, 1998b*; *Morozov, 2000*), did not emerge as genomically distinct from other conspecific populations and are likely to represent ecomorphological adaptations controlled by few genes. Sample NBME 1039630, which had been labeled as *N. tenuirostris*, and sample MCZR 15733, which was initially identified as an *N. arquata* that shares many morphological features with *N. tenuirostris*, clustered with *N. arquata* samples (*Figure 1—figure supplement 1D*; *Supplementary file 1*). Both samples were assigned to *N. arquata* in subsequent phylogenetic analyses.

## Sequence assembly

For sequence-based analyses, reads were assembled using HybPiper 1.3.1 (*Johnson et al., 2016*) (reads_first) to yield sequence loci. Firstly, reads were mapped to the target sequences using BWA 0.7.17 (*Li and Durbin, 2009*) and sorted by gene. Contigs were then assembled from the reads mapped to respective loci using SPAdes 3.13 (*Bankevich et al., 2012*) with a coverage cutoff value of 20. Using Exonerate 2.4.0 (*Slater and Birney, 2005*), these contigs were then aligned to the target sequences and sorted before one contig per locus was chosen to yield the final sequences. We inspected locus lengths (get_seq_lengths) and recovery efficiency (hybpiper_stats) across all loci. We then investigated potentially paralogous loci (paralog_investigator) by building gene trees using FastTree 2.1.11 (*Price et al., 2010*) (paralog_retriever), leading to the removal of 10 loci. Finally, sequences from the same loci were retrieved from all samples to generate a multisequence alignment for each locus (retrieve_sequences.py). All loci retained were present in at least 80% of individuals and constituted at least 60% of the length of total target loci. In summary, a total of 525 loci with a mean length of 969 bp (492–6,054 bp) were recovered from 62 samples.

## Phylogenomic analyses using sequence data

Multisequence alignment was performed for each locus using MAFFT 7.470 (*Katoh and Standley, 2013*), allowing for reverse complement sequences as necessary. Alignments were checked for gaps using a custom script, and loci with >35% gaps were removed from downstream analyses. A total alignment length of 514,771 bp was obtained.

Phylogenomic analyses were performed on a concatenated dataset as well as on individual gene trees. Concatenation was performed with abioscript 0.9.4 (*Larsson, 2010*) (seqConCat). For the concatenated dataset, we constructed maximum-likelihood (ML) trees using RAxML 8.2.12 (*Stamatakis, 2014*) with 100 alternative runs on distinct starting trees. We applied the general time reversible

substitution model with gamma-distributed rate variation among sites and with the estimation of the proportion of invariable sites (GTR + I + G) (*Abadi et al., 2019*; *Arenas, 2015*).

For individual gene trees, the best substitution model for each locus was determined using jModelTest 2.1.10 (*Darriba et al., 2012*) by virtue of the corrected Akaike information criterion value. We then constructed ML trees in PhyML 3.1 with the subtree pruning and regrafting algorithm, using 20 initial random trees. We performed 100 bootstrap replicates with ML estimates for both proportions of invariable sites and the value of the gamma shape parameter. Individual gene trees were then rooted with Newick Utilities 1.3.0 (*Junier and Zdobnov, 2010*). We removed one locus from downstream analyses due to the absence of an outgroup sequence such that 524 loci were retained across 62 samples.

Species tree analyses were performed using the rooted gene trees in MP-EST 1.6 (*Liu et al., 2010*), without calculation of triple distance among trees. We grouped samples by species and performed three runs of 10 independent tree searches per dataset (*Cloutier et al., 2019*). To calculate the bootstrap values of the species tree, we performed multi-locus, site-only resampling (*Mirarab, 2014*) from the bootstrap trees' (100 per gene) output from PhyML. The resulting 100 files, each with 100 bootstrap trees, were rooted and species tree analyses were performed in the same manner for each file in MP-EST. The best tree from each run was identified by the best ML score and compiled. Finally, we used the majority rule in PHYLIP 3.695 (*Felsenstein, 2009*) to count the number of times a group descending from each node occurred so as to derive the bootstrap value (consense).

For the estimation of divergence times, we applied MCMCtree and BASEML (*dos Reis and Yang, 2011*), a package in PAML 4.9e (*Yang, 2007*). To prepare the molecular data from 62 samples and 524 loci, we compiled the DNA sequence of each sample and combined all samples onto separate rows of the same file. We then obtained consensus sequences for each species using Geneious Prime 2020.2 (*Kearse et al., 2012*), with a majority support threshold of 50% and ignoring gaps. We visually checked the resulting consensus sequences to ensure that ambiguous bases remained infrequent. Consensus sequences were organized by loci as per the input format for MCMCtree. We then prepared the input phylogenetic tree using the topology estimated in MP-EST with calibrations of the two most basal nodes, namely between our outgroup (*Tringa totanus*) and all *Numenius* species, as well as that between the whimbrel and curlew clades within *Numenius*. Due to a lack of known fossils within the genus *Numenius*, we were unable to perform fossil node calibrations. Instead, we utilized p-distance values calculated from the COI sequences of *Numenius* species. Specifically, we applied the bird COI mutation rate of 1.8% per million years (*Lavinia et al., 2016*) and converted mean, maximum, and minimum p-distance values of both nodes to time (100 million years ago (MYA)). We maintained a conservative position and scaled the COI-based timings by a factor of two to obtain the final lower and upper bounds of node timings. We used the default probability of 0.025 that the true node age is outside the calibration provided.

To run MCMCtree, we first calculated the gradient and Hessian matrix of the branch lengths with the GTR substitution model applied, using default values of gamma rates and numbers of categories (mcmctree-outBV.ctl). We then performed two independent Markov chain Monte Carlo (MCMC) samplings of the posterior distribution of divergence times and rates (mcmctree.ctl). All default values were used except that a constraint on the root age was set to <0.3 (100 MYA). We also varied the prior for the birth-death process with species sampling and ensured that time estimates are not affected by the priors applied *Dos and Yang, 2019*. We then performed convergence diagnostics for both runs in R to ensure that posterior means are similar among multiple runs, while checking that the parameter space has been explored thoroughly by the MCMC chain. Finally, we conducted MCMC sampling from the prior with no data to check the validity of priors used by comparing them with the posterior times estimated. Again, two independent MCMC samplings were performed with convergence diagnostics.

Phylogenetic trees were visualized in FigTree 1.4.4 (*Rambaut, 2018*) with bootstrap values and node ages (MYA) including the 95% credibility intervals. Evolutionary distinctness and phylogenetic diversity were calculated for each branch (*Jetz et al., 2014*) using the divergence times estimated in MCMCTree.

## Demographic history reconstruction

We derived trends in effective population size using stairway plot 2.1.1, which uses the SNP frequency spectrum and is suitable for reduced representation datasets (*Liu and Fu, 2020*; *Patton et al., 2019*). From the dedicated SNP sets that were created without minor allele frequency cut-off, we calculated a folded site frequency spectrum using vcf2sfs.py 1.1 (*Marques et al., 2019*). We assumed a mutation rate per site per generation of 8.11 $e^{-8}$, as estimated for shorebirds in the same order as *Numenius* (Charadriiformes) (*Wang et al., 2019*), and applied the following generation times respectively: *N. americanus* 7 years, *N. arquata* 10 years, *N. hudsonicus* 6 years, *N. phaeopus* 6 years, *N. tenuirostris* 5 years (*Bird et al., 2020*; *IUCN, 2020*). We ran a stairway plot on all species, applying the recommended parameters.

Stairway plot is expected to perform at its highest accuracy in the reconstruction of demographic history in the recent rather than distant past. However, the definition of the recent past varies from anywhere between 30 generations to ~40,000 generations before the present (*Liu and Fu, 2015*; *Patton et al., 2019*). We did not set a cutoff for the time period investigated but let it be determined by the program itself. Additionally, we omitted reconstructions of the last 10 steps to avoid overinterpretation of the distant past (*Liu and Fu, 2015*). We only displayed the results from the time period for which there was data across all species, and only for four species represented by five or more samples (stairway_plot_es Stairbuilder), as recommended for accurate inference (X. Liu, personal communication, October 14, 2020). We later also included *N. americanus*, for which we had four samples, as its sample size did not appear to affect the reliability of the results (*Figure 1*). We were unable to include the remaining species (*N. borealis*, *N. tahitiensis,* and *N. minutus*) as their demographic history reconstructions were clearly affected by a lack of sufficient sample size. For *N. borealis,* two out of the five samples showed high missingness, with adverse effects on stairway plot analyses, both in runs including all five samples and those that excluded the two samples of high missingness (*Figure 1— figure supplement 2*). Our ability to trial a large number of samples for laboratory work was also limited by the availability of target enrichment baits.

We attempted to infer demographic history using sequentially Markovian coalescent-based methods, which are more reliable for older timescales, to corroborate our stairway plot results (*Patton et al., 2019*). In particular, we used the Pairwise Sequentially Markovian Coalescent (PSMC) model (*Li and Durbin, 2011*) as it has been successfully applied to reduced-representation datasets (*Liu and Hansen, 2017*). This method allows for analyses of all species as only one sample per species is required as input. However, given the constraints created by the sampling density of our target enrichment dataset, we were unable to run PSMC successfully.

## Ecological niche modeling

We performed ecological niche modeling (*Anderson et al., 2011*) to predict the extent of suitable breeding areas for species across the duration of our demographic history reconstruction. We were able to do so for each species in the stairway plot except *Numenius tenuirostris* due to the paucity of confirmed breeding records. We obtained species occurrence data from *eBird, 2021* and the Global Biodiversity Information Facility (GBIF; using only records with coordinate uncertainty <1,000 m) (*GBIF.org, 2022a*; *GBIF.org, 2022b*; *GBIF.org, 2022c*; *GBIF.org, 2022d*; *GBIF.org, 2022e*; *GBIF.org, 2022f*). For *N. phaeopus*, we also included confirmed breeding localities from *Lappo et al., 2012* to improve the sample size. Species occurrence data from various sources were combined and further filtered (*Supplementary file 3A*). Occurrence points were filtered by month to retain only records in peak breeding months of respective species (*Birds of the World, 2022*). For species with sufficient occurrence points, occurrence points were also filtered by year to match the time range of the climatic variables, i.e., 1960–1990. Otherwise, occurrence records from all years were used to maximize sample size. For species that span the entire Palaearctic (*N. phaeopus* and *N. arquata*), sampling density was much higher in Europe. To account for the extreme sampling bias, in addition to generating a kernel density estimate (see next paragraph), occurrence records within Europe for these two species were randomly down-sampled to match sampling density across the rest of the Palearctic. Occurrence records outside of the known breeding area of each species were removed (*BirdLife International and Handbook of the Birds of the World, 2017*; *Lappo et al., 2012*). Finally, to reduce spatial autocorrelation, occurrence records were thinned using a 50 km buffer (*Aiello-Lammens et al., 2015*).

To account for sampling bias specific to shorebirds, such as those of this study, we generated a kernel density estimate using the R package spatialEco 1.3–7 (*Evans, 2021*) based on the occurrences of species within Scolopacidae. The kernel density estimates were then used to inform background point selection (i.e. matching sampling bias) (*Kramer-Schadt et al., 2013*). For each species, we further limited the sampling of background points to areas outside a 10 km buffer around occurrence points and within a 500 km buffer around the known breeding area using the R packages terra 1.5–21 and raster 3.5–15 (*Hijmans, 2022b*; *Hijmans, 2022a*). A total of 10,000 background points were then sampled without replacement for each species.

All 19 bioclimatic variables (raster; 2.5 arcmin resolution of ~4.5 km) from WorldClim 1.4 (*Hijmans et al., 2005*) were obtained for the present-day (1960–1990), mid-Holocene (6,000 years ago), and LGM (22,000 years ago). Bioclimatic variables were then prepared for input into Maxent 3.4.4 using QGIS 3.4 *QGIS.org, 2022* following *De Alban, 2022*. Polygon shapefiles were first created for each species, which included the present-day breeding distribution as well as areas south of that to accommodate for potential shifts in distribution around the LGM. These polygons were then used to crop the bioclimatic variable raster for each respective species (*Conrad et al., 2015*).

We applied Maxent 3.4.4, which makes use of presence-only data and environmental data to model species' geographical distributions (*Phillips et al., 2006*). Species-specific Maxent analyses were performed using the respective breeding occurrence records, background points, and present-day bioclimatic variables of each species. To reduce collinearity among predictors, we removed predictors with a high variance inflation factor (>3) for each species. To facilitate parameter tuning, 20 candidate models were built for each species and evaluated using the R package ENMeval 2.0.3, testing combinations of feature classes (L, LQ, LQH, LQPH) and regularisation multipliers (0.5, 1, 2, 3, 4) (*Kass et al., 2021*; *Merow et al., 2013*). To test for model overfitting and transferability, candidate models were cross-validated using the 'block' partitioning technique (i.e. occurrences and background points were partitioned into four spatial blocks, where occurrence numbers among partitions are equal) (*Fourcade et al., 2018*; *Muscarella et al., 2014*). Candidate models with omission rates (minimum training presence threshold) exceeding 0.2 were rejected. The candidate model with the highest area under the receiver-operator curve (AUC) was selected as the final model (*Supplementary file 3B*) and used to predict suitable breeding areas under present-day, mid-Holocene, and LGM climate conditions (*Figure 1—figure supplement 3*).

Predicted species distributions were visualised in R (*R Development Core Team, 2022*). We performed a binary classification of predicted occurrence probability using the maximum sum of sensitivity plus specificity threshold (*Liu et al., 2013*) and calculated suitable breeding area using the R package raster 3.5–15.

## Acknowledgements

We thank the following personnel and institutions for their generous contribution of samples (*Supplementary file 1*): Paul Sweet and Thomas Trombone at the American Museum of Natural History (AMNH, New York); Robert Palmer and Leo Joseph at the Australian National Wildlife Collection (ANWC, Canberra); Molly Hagemann at the Bernice Pauahi Bishop Museum (BPBM, Hawaii); David Allan at the Durban Natural Science Museum (DNSM, Durban) and Celine Santillan who assisted in sample transport; Ben Marks at the Field Museum of Natural History (FMNH, Chicago); Foo Maosheng at the Lee Kong Chian Natural History Museum (LKCNHM, Singapore); Carla Marangoni and Gloria Svampa at the Museo Civico di Zoologia (MCZR, Rome); Henry McGhie at the University of Manchester, Manchester Museum (MMUM, Manchester); Robert Prŷs-Jones, Mark Adams, Alex Bond, Ari Benucci and Douglas Russell at the Natural History Museum, London (NHMUK, Tring); Manuel Schweizer at the Naturhistorisches Museum der Bürgergemeinde Bern (NMBE, Bern); Bob McGowan at the Natural Museum of Scotland (NMS, Edinburgh); Joanna Sumner at Museums Victoria (NMV, Melbourne); Angela Ross at the National Museums NI (NMNI, Northern Ireland) and David Allen and Graeme Buchanan who assisted in sample transport; Jan Bolding Kristensen at the Natural History Museum of Denmark (SNM, Copenhagen); José Alves and Camilo Carneiro at the University of Iceland (UOI, Reykjavik); Sharon Birks at the Burke Museum, University of Washington (UWBM, Seattle); Pavel S Tomkovich, Dmitry Shitikov and Vladimir Sotnikov at the Zoological Museum of Moscow State University (ZMMU, Moscow); and Fyodor Kondrashov and Lisa Chilton who assisted in sample transport. Fletcher Smith assisted in Mozambique (with permission from

Lucilia Chuquela, Museu de História Natural and Universidade Eduardo Mondlane, Maputo) with further assistance from Rebecca and Cyril Kormos, Patricia Zurita and Vinayagan Dhamarajah. HZT acknowledges Elize Ying Xin Ng, Pratibha Baveja, Yong Chee Keita Sin, Shivaram Rasu, Dominic Yong Jie Ng, Meng Yue Wu, Liu Xiaoming, and Jose Don De Alban for assistance with laboratory procedures and analyses. The authors acknowledge support from the National Genomics Infrastructure in Stockholm funded by Science for Life Laboratory, the Knut and Alice Wallenberg Foundation and the Swedish Research Council, and SNIC/Uppsala Multidisciplinary Center for Advanced Computational Science for assistance with massively parallel sequencing and access to the UPPMAX computational infrastructure.

## Additional information

### Funding

| Funder | Grant reference number | Author |
| --- | --- | --- |
| Swedish Research Council | 2019-03900 | Martin Irestedt |
| DBT-Ramalingaswami Fellowship | BT/HRD/35/02/2006 | Kritika M Garg |
| South East Asian Biodiversity Genomics (SEABIG) Grant | WBS R-154-000-648-646 | Balaji Chattopadhyay |
| South East Asian Biodiversity Genomics (SEABIG) Grant | WBS R-154-000-648-733 | Balaji Chattopadhyay |
| Trivedi School of Biosciences, Ashoka University | | Balaji Chattopadhyay |
| Singapore Ministry of Education Tier 2 grant | WBS R-154-000-C41-112 | Frank E Rheindt |

The funders had no role in study design, data collection and interpretation, or the decision to submit the work for publication.

### Author contributions

Hui Zhen Tan, Conceptualization, Resources, Data curation, Formal analysis, Investigation, Visualization, Methodology, Writing - original draft, Project administration, Writing – review and editing; Justin JFJ Jansen, Gary A Allport, Conceptualization, Resources, Writing – review and editing; Kritika M Garg, Balaji Chattopadhyay, Resources, Supervision, Methodology, Writing – review and editing; Martin Irestedt, Resources, Funding acquisition, Writing – review and editing; Sean EH Pang, Formal analysis, Methodology, Writing – review and editing; Glen Chilton, Resources, Writing – review and editing; Chyi Yin Gwee, Supervision, Methodology, Writing – review and editing; Frank E Rheindt, Conceptualization, Resources, Supervision, Funding acquisition, Visualization, Methodology, Writing - original draft, Project administration, Writing – review and editing

### Author ORCIDs

Hui Zhen Tan http://orcid.org/0000-0002-6750-7506
Gary A Allport http://orcid.org/0000-0003-2261-6386
Kritika M Garg http://orcid.org/0000-0003-3510-3408
Chyi Yin Gwee http://orcid.org/0000-0003-1706-0520
Frank E Rheindt http://orcid.org/0000-0001-8946-7085

### Decision letter and Author response

Decision letter https://doi.org/10.7554/eLife.85422.sa1
Author response https://doi.org/10.7554/eLife.85422.sa2

## Additional files

### Supplementary files
• Supplementary file 1. Sampling information. Legend: Details of samples collected for this study.

• Supplementary file 2. Evolutionary distinctness of *Numenius* species. Legend: Evolutionary distinctness, phylogenetic diversity, and evolutionarily distinct and globally endangered (EDGE) scores of *Numenius* species.

• Supplementary file 3. Ecological niche modeling information. Legend: Details of occurrence points, parameters, and results of ecological niche modeling using Maxent.

• MDAR checklist

### Data availability
DNA reads generated in this study are available on Sequence Read Archive under BioProject PRJNA742889. The reference genome generated in this study is available at DDBJ/ENA/GenBank as a Whole Genome Shotgun project under the accession JARKVS000000000. The version described in this paper is version JARKVS010000000. Pipelines and analysis codes are available on GitHub: https://github.com/tanhuizhen/Numenius_Target-enrichment_Analyses (copy archived at *Tan, 2023*).

The following datasets were generated:

| Author(s) | Year | Dataset title | Dataset URL | Database and Identifier |
|---|---|---|---|---|
| Tan HZ, Jansen JFJ, Allport GA, Garg KM, Chattopadhyay B, Irestedt M, Pang SEH, Chilton G, Gwee CY, Rheindt FE | 2023 | *Numenius* target enrichment libraries | https://www.ncbi.nlm.nih.gov/bioproject/PRJNA742889 | NCBI BioProject, PRJNA742889 |
| Tan HZ, Jansen JFJ, Allport GA, Garg KM, Chattopadhyay B, Irestedt M, Pang SEH, Chilton G, Gwee CY, Rheindt FE | 2023 | *Numenius phaeopus* reference genome | https://www.ncbi.nlm.nih.gov/nuccore/JARKVS000000000 | NCBI GenBank, JARKVS000000000 |

The following previously published datasets were used:

| Author(s) | Year | Dataset title | Dataset URL | Database and Identifier |
|---|---|---|---|---|
| Project Vertebrate Genomes | 2019 | Taeniopygia guttata (zebra finch) genome sequencing and assembly, primary haplotype | https://www.ncbi.nlm.nih.gov/assembly/GCA_003957565.1 | NCBI Assembly, GCA_003957565.1 |
| University Uppsala | 2013 | Ficedula albicollis Genome sequencing and assembly | https://www.ncbi.nlm.nih.gov/assembly/GCA_000247815.2 | NCBI Assembly, GCA_000247815.2 |
| Küpper et al. | 2015 | Genome assembly of the ruff (Philomachus pugnax) | https://www.ncbi.nlm.nih.gov/datasets/genome/GCA_001458055.1/ | NCBI Assembly, GCA_001458055.1/ |

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
