## [Editor Report]

This study uses genomic inferences to reconstruct past population sizes of whimbrel and curlew shorebirds, along with niche modeling approaches, to explore changes in those populations over millenia. Steppe-dependent breeding species appear to have declined more prominently than species that breed in other habitats. The coincident timing of these declines of steppe-dependent breeding shorebirds, and the extinction of the mammalian megafauna that likely maintained that habitat, raises the intriguing possibility that those mammalian extinctions had broad effects on these shorebirds and the entire community of steppe-dependent organisms.

---

## [Decision Letter]

**Decision letter after peer review:**

[Editors’ note: the authors submitted for reconsideration following the decision after peer review. What follows is the decision letter after the first round of review.]

Thank you for submitting the article "Historical climate change and megafaunal extinctions linked to genetic diversity declines in shorebirds" for consideration by *eLife*.

Your article has been reviewed by three peer reviewers, and the evaluation has been overseen by Irby Lovette as the Reviewing Editor, and a Senior Editor. The reviewers have opted to remain anonymous.

Comments to the Authors:

We are sorry to say that, after consultation with the reviewers, we have decided that your article is not suitable in its present form for publication in *eLife*. That said, we would be willing to consider a fresh submission of a thoroughly revised version that includes additional data and carefully addresses the reviewers' other concerns.

All reviewers were intrigued by your study and found much to admire in it. The main limitations are those that are clearly identified in the reports below, namely: (1) the fact that the Ne analyses are available for only a subset of the species of interest; and (2) concerns that the presentation of these results extrapolates beyond the available data in assigning causality to megafaunal effects on broadscale habitat changes. Our sense is that these two critiques are related, in that having a small sample size of demographic reconstructions makes it hard to test rigorously for a temporal association of the type predicted by the megafaunal hypothesis.

The Guest Reviewing Editor notes that there are three N. borealis specimens at Cornell, and that they would be happy to sample them for you and facilitate a loan, should you want to use them to augment your sample size of that extinct taxon.

*Reviewer #1:*

Overall I am very enthusiastic about this study, which has a high degree of novelty. In particular I have never before heard of the potential link between megafaunal extinction around the time of the LGM, broadscale habitat change as a direct consequence, and population declines in steppe-breeding shorebird species. Some of the links in this chain of causality will be hard to test rigorously, but the hypothesis is fascinating and the demographic data summarized in this submission are consistent with this pattern.

The data and analysis methods seem generally robust and appropriate. The phylogenetic analyses could have been done in many ways (and those of us from this discipline love to quibble about these nuances), but the resulting tree here seems robust in terms of topology and underlying data depth. The phylogeny includes some interesting new information on the affinities of these species to one another.

For me, the real meat of this paper is in the stairway plots of historical Ne, as summarized in Figure 1C. The four species included there each have a distinctively different pattern of Ne over time. Since these analyses and estimates are so integral to the paper and its inferential conclusions, it may be worthwhile to explore/address whether they have relevant biases or limitations. For example, Patton et al. (2019) (Contemporary Demographic Reconstruction Methods Are Robust to Genome Assembly Quality: A Case Study in Tasmanian Devils) report that these particular methods are most reliable at estimating only fairly recent (30 generations) Ne's.

My primary critique of this submission is that the historical demographic estimates are made for less than half (4 of 9) of these shorebird species. The within-species sample size cutoff of n=5 for these estimates means that several additional species could have been included if just one more sample was available. I understand the difficulty of sourcing material for some of these species, but Ne stairway plots for more taxa would substantially elevate the inferential power of this cross-taxon comparison in which it seems that tundra-breeding species might have a different pattern than temperate-breeding species.

With a greater sample of species to work with, it might then be possible to formally test whether species with the largest predicted shifts in breeding habitat show corresponding changes in estimated Ne. Such a result would substantially bolster the inferential argument presented here about that potential relationship.

In terms of presentation, the paper jumps between historical climate change at recent geological time scales, and current anthropogenic climate change. The two extinct curlew species in this clade almost certainly went extinct owing to primary causes other than anthropogenic climate change. Past effects of climate and broad-scale habitat change are certainly relevant, but the chain of causality presented here could be improved.

Some statements about past processes are presented as facts, whereas my understanding is that they are still subject to debate. In particular the causality of human hunting>megafaunal extinction>widespread habitat change from grasslands to forest is more of a hypothesis. Similarly, the evidence for lower genetic diversity creating higher extinction risk is fairly nuanced. In presenting their own results, the authors may want to be slightly less declarative about causality, such as in saying "Our study revealed that climatic and environmental changes impacted the genetic diversity of curlews" when the data are corelative and based on only a couple of taxa.

In general I recommend focusing more tightly on the inferences about the potential relationship between past Ne and broadscale habitat change during the Pleistocene and thereafter. These are fascinating and intriguing enough on their own, but they are indeed highly inferential. In terms of present and future risk, anthropogenic climate change is probably not among the greatest potential drivers of population declines and possible extinction in this group of shorebirds.

Recommendations for the authors:

I recommend substantially reworking the abstract to make it less general and to highlight even more the truly novel components of this study.

If you decide to add more genomic data, we have two borealis specimens in the Cornell Museum of Vertebrates that I would be happy to sample for you (though I do understand the challenges of adding data, doing the reanalyses, and arranging all of the permits to get samples from here to there…). I really do wish that you could add at least borealis and americanus to the stairway plots, since they each typify one of the different habitats in your general hypothesis.

I'm not sure that this is really one of the MOST 'extinction prone' groups of birds, especially in comparison to various island groups.

Personally I think that the timing of human spread around the globe and megafaunal extinction is too close to be spurious, but that debate goes back decades and I'm not sure there is consensus among the experts on whether it was causal?

*Reviewer #2:*

In this paper, the authors use genomic data to explore how historical processes may have helped shape the current conservation statuses of a group of threatened (and in some cases extinct) migratory birds. They conclude that nearly all of the species in the group have exhibited declines in their effective population size since the Last Glacial Maxima, but that these declines have been particularly large in more temperate breeding species. The authors then attribute the more substantial declines of the temperate breeding species to the larger overlap these species have had with historic anthropogenic activities.

This paper has a number of strengths: (1) It is clearly written and easy to follow. (2) It provides the first species-level phylogeny of this group of species. (3) It focuses on a group of species that has already experienced two extinctions and includes a number of other threatened and endangered taxa; as such insights that may contribute to their conservation are sorely needed.

The paper also has a number of significant weaknesses. Most importantly: (1) No testable hypotheses or predictions are set forth. (2) While sampling across taxon is extensive, sampling within each taxa is more limited and precludes the inclusion of most in the core analyses. (3) No effort is made to reconstruct the historic ranges of any of the species or, therefore, to robustly analyze the historical processes that may have been most likely to affect their effective population sizes.

In the end, I find it difficult to know what conclusions to walk away with. With only five taxa meeting the threshold of five individuals for inclusion in the generation of effective population size 'histories' it's really difficult to run any formal analyses that might allow for robust statistical tests. This then limits the ability of the authors to draw strong conclusions about which groups of species have experienced the strongest declines in genetic diversity.

I also urge the authors to read and digest Crisp et al. 2011 in Trends in Ecology and Evolution. That paper provides a framework for how biogeographical studies can generate testable hypotheses, something that this manuscript lacks. Besides 'eyeballing' Figure 1C, it's impossible to assess whether any of the historical processes discussed actually were likely to play a role in the observed declines in effective population size. What alternative hypotheses might there be that could also be tested and refuted?

*Reviewer #3:*

Overall, a study such as the one presented is important for understanding our current biodiversity crisis and I appreciate the work. The methods are thorough and very nicely written. At this stage based on the framework presented the authors do not satisfactorily present evidence that megafauna decline and climate change drove the reduced population sizes in the focal species. To strengthen the work there need to be testable hypotheses presented and I would recommend moving away from the idea that only the glacial-interglacial end of Pleistocene climate change event is responsible for extinction while ignoring long term human impacts on biodiversity across the entirety of the Holocene (past 11,000 years).

The authors use genetic data to evaluate the evolutionary relationships and the population size of species of Numenius (shorebird) species. They found a decline in population sizes starting at the end of the last glacial of the Pleistocene. These declines are comparable to the declines found in megafauna which generally (on continents) occurred at toward the end of the Pleistocene and the end of the last glacial. However, there is not a direct link demonstrated that the declines of these shorebirds were caused by the declines of megafauna (which is what the title states). While many large mammals were ecosystem engineers, and the loss of these species could possibly impact the breeding habitats of these species leading to a reduction of breeding grounds resulting in declines this conjecture there is no data supporting these linkages herein. For example, one could also argue that increased human population size could also be driving these patterns or just the loss of wintering habitat due to higher sea levels and reduced coastline in some areas may be driving these patterns as well. I would argue that you have a more evidence that (as stated in the abstract): "Species breeding in temperate regions, where they widely overlap with human populations, have been most strongly affected". This falls in line with the fact that Eskimo Curlew was decimated by human hunting in the 19th century and was already in decline from the conversion of breeding ground habitat to agricultural land. Human activities since the late Pleistocene (which includes landscape modifications) have led to the current extinction crisis that has occurred since the late Pleistocene. It would be more effective to highlight this more than megafaunal losses driving the population declines in these species. Trophic cascades driving extinction/population reductions are a possibility but without direct evidence it is best left for discussion points.

To significantly strengthen the argument that climate change is the driving factor of declines I recommend investigating how effective population sizes changed across the Quaternary (past 2.6 million years). The last Pleistocene glacial- modern (Holocene) interglacial climate change event is not a unique and these dynamic conditions have occurred 20+ times across the Quaternary. Therefore, if you find that multiple times across the past 2.6 million years there has been a reduction in effective population size that corresponds to interglacial intervals then perhaps you can state that climate change has led to the decline of Numenius populations. Therefore, only looking at a single time point where there were much more destructive processing taking place i.e., human direct and indirect impacts, I find it is hard to draw this firm conclusion that it is climate change driving these patterns in your data. To better sort out various processes and patterns I would recommend restructuring the introduction to provide the background regarding the history of the Quaternary from a climate, extinction, and human perspective and provide testable hypotheses e.g., 1. Across the past 2.6 million years during interglacial intervals, when, presumably, breeding habitats and shoreline (wintering habitats) were less extensive, effective population sizes were greatly reduced due to limited resources. If this finding is not the case, then you would need another factor to have driven these species to have reduced population sizes at the end of the Pleistocene-Holocene and throughout the Holocene.

Overall, human impacts especially highlighting what happened to the Eskimo Curlew as hunting (loss of millions of individuals), breeding habitat destruction for agriculture, and the extinction of its main food item on the breeding ground (Rocky Mountain locust) should be included to strengthen the work. You can even provide human population size estimates, as available, in various parts of the breeding ranges across the Holocene.

I appreciate this project and manuscript and the acknowledgement that we need to understand declines and extinction of species considering the current diversity crisis.

Recommendations for the authors:

1. Title is too general for the study and the results.

2. The climate change event that occurred at the end of the Pleistocene and beginning of the Holocene was not a unique event. These glacial-interglacial intervals occurred 20+ times across the past 2.6 million years. Accordingly, it is important to evaluate whether there were multiple reductions of effective population size in these species across this time. This would strengthen the argument that climate change is impactful on these shorebirds' distributions and population size. If not then another factors need to be evaluated.

Line 107 – Is the decline seen in differential hunting of various species after 700 years ago?

Figure 1 caption – It looks like the demographic modeling wasn't performed for more than half of your focal species. This should be explicitly stated in the methods/results/discussion for transparency.

Line 150 in the figure caption there is a missing space

For the historical specimen DNA extraction methods please briefly elaborate and provide details on how you modified the Qiagen kit for the historical specimen extractions. Currently there is a citation but please briefly include some extra details so folks can easily access them.

Line 263 – is the perl script for removing potential linked SNPs available somewhere or upon request? Might be worthwhile mentioning that it is to strengthen the methods.

[Editors’ note: further revisions were suggested prior to acceptance, as described below.]

Thank you for resubmitting your work entitled "Megafaunal extinctions-not climate change-seem to explain Holocene genetic diversity declines in Numenius shorebirds" for further consideration by *eLife*. Your revised article has been evaluated by Christian Rutz (Senior Editor) and a Reviewing Editor.

The manuscript has been improved but there are some remaining issues that need to be addressed, as outlined below:

Essential revisions:

As the reviewing editor, I am supportive of the publication of this paper in its current form. However, Reviewer 1 still has reservations about the power of inference here for inferring a causal effect of the megafaunal extinctions. The way *eLife* works, we have had a dialogue about these disconnects, and even after that back-and-forth we are in different places about the inferential power of this study. To reconcile these different levels of comfort with that inference, please consider the following small changes:

1) in the title, change "seem to" to "may"

2) add an additional qualifier to the abstract to indicate that the megafaunal scenario is one of multiple potential explanations.

I can't speak for the reviewer, but I come from this partly from a place of thinking that the megafaunal hypothesis is really cool and that it is indeed consistent with your available data. I think that highlighting it here might well spark other investigations of this general type and that this novelty is worth something of its own. I also come from a comparative phylogeography background which helps me understand the limitations of the N that evolution has provided to test any particular hypothesis, along with the awareness that it is very rare to have all historical data points converge on a simple scenario.

*Reviewer #1 (Recommendations for the authors):*

I commend the authors on their revised effort and greatly appreciate the addition of the ecological niche modeling, as well as explicit hypotheses to be tested. To my eyes the modeling appears robust. The paper is also clearly written and flows nicely from idea to idea.

Where I continue to struggle, however, is with the inferential power of the study. I apologize, but I cannot quite move past the fact that, with only five species, it is impossible to run any statistical analyses linking changes in Ne with changes in habitat availability or any other historical factor.

– For instance, Reviewer #3 brought up in the previous round of comments the possibility that rising sea levels reduced stopover/nonbreeding habitat simultaneous with the megafaunal extinction. That possibility is not addressed with any analyses, and quite possibly can't be addressed given the limited sample sizes involved, but it should nonetheless be mentioned.

– What is more, the apparent disconnect between the ecological niche modeling results and Ne results, gives me pause. While the reason offered is plausible, it's hard not to get stuck there without some sort of actual analysis to help explain the difference.

– Finally, I am struck by the dramatic differences in Ne patterns between N. phaeopus and N. hudsonicus given their biological and ecological similarity. Is this an artifact of only having 5 N. hudsonicus? (I could provide you with dozens of samples if you would like!) In either case, some mention of this discrepancy in patterns should at least be mentioned in the main text.

Thus, all told, I just feel uncomfortable pinning it all on the megafaunal extinction and urge caution with framing the paper (including the title) that way. I know how hard it is to pin contemporary declines on any single factor and, so, it seems that it is likely even harder to do so when looking into the past thousands of years.

---

## [Author Response]

[Editors’ note: the authors resubmitted a revised version of the paper for consideration. What follows is the authors’ response to the first round of review.]

Reviewer #1:Overall I am very enthusiastic about this study, which has a high degree of novelty. In particular I have never before heard of the potential link between megafaunal extinction around the time of the LGM, broadscale habitat change as a direct consequence, and population declines in steppe-breeding shorebird species. Some of the links in this chain of causality will be hard to test rigorously, but the hypothesis is fascinating and the demographic data summarized in this submission are consistent with this pattern.The data and analysis methods seem generally robust and appropriate. The phylogenetic analyses could have been done in many ways (and those of us from this discipline love to quibble about these nuances), but the resulting tree here seems robust in terms of topology and underlying data depth. The phylogeny includes some interesting new information on the affinities of these species to one another.

We thank the reviewer for the validation of our novel research questions and robust methods.

For me, the real meat of this paper is in the stairway plots of historical Ne, as summarized in Figure 1C. The four species included there each have a distinctively different pattern of Ne over time. Since these analyses and estimates are so integral to the paper and its inferential conclusions, it may be worthwhile to explore/address whether they have relevant biases or limitations. For example, Patton et al. (2019) (Contemporary Demographic Reconstruction Methods Are Robust to Genome Assembly Quality: A Case Study in Tasmanian Devils) report that these particular methods are most reliable at estimating only fairly recent (30 generations) Ne's.

We appreciate the reviewer’s feedback regarding our stairway plot analyses. In “Materials and methods: Demographic history reconstruction”, we have now provided detailed elaborations on the rationale for stairway plot being our method of choice for demographic history reconstructions. Specifically, the new version of our manuscript acknowledges the utility of stairway plots for reduced representation datasets and recent time scales. As the reviewer mentioned, Patton et al. (2019) found that stairway plot is most reliable within 30 generations before present (<100 years). On the other hand, Liu and Fu (2015) found that despite higher dispersion, the mean results of stairway plot reliably track known demographic histories even up to 1 mya (~40,000 *Homo sapiens* generations). The time period for which stairway plot is informative seems to vary across papers (Nadachowska-Brzyska, Konczal, and Babik, 2022). When running stairway plot, we did not restrict the time range for reconstructions but allowed the program to determine the suitable upper limit. We also refrained from discussions of results from the last ten steps of the resulting stairway plot, as recommended by Liu and Fu (2015). All these details can now be found in the Methods section of the revised manuscript.

My primary critique of this submission is that the historical demographic estimates are made for less than half (4 of 9) of these shorebird species. The within-species sample size cutoff of n=5 for these estimates means that several additional species could have been included if just one more sample was available. I understand the difficulty of sourcing material for some of these species, but Ne stairway plots for more taxa would substantially elevate the inferential power of this cross-taxon comparison in which it seems that tundra-breeding species might have a different pattern than temperate-breeding species.

We agree with the reviewer that it is unfortunate that the loss of historic museum samples due to low DNA quality has prevented us from including even more species in our demographic reconstruction. We have pursued three avenues to address the reviewer’s concerns:

1) Following the reviewer’s encouragement, we were eventually able to perform stairway plot analyses for one additional species (*N. americanus*), raising the total number of species analysed to 5 out of 9 species. With this addition, our stairway plot analysis now features representatives of tundra and temperate-breeding species from both North America and Northern Eurasia, strengthening our cross-taxon comparisons.

2) We additionally ran multiple stairway plots for one more crucial species, the extinct Eskimo Curlew, under different sampling regimes (see new Figure S2). While our total sample size for Eskimo Curlews was n=5, theoretically allowing us to include this species, two of the samples were characterized by extremely poor DNA quality. We ran stairway plot for all 5 individuals, including the two degraded samples, as well as for 3 out of 5 (minus the two degraded samples). In both cases, the resultant stairway plot is clearly biased, either by low sample size or low input data quality. Similar attempts have also been made for all other species that were excluded from stairway plot analyses due to low sample size. While not adding considerably to our power of inference, we present these results in the Supplement.

3) Beyond stairway plot, we have also implemented the Pairwise Sequentially Markovian Coalescent (PSMC) model, a complementary analysis that is suitable for low sample sizes, in an attempt to include the species excluded from stairway plots. While the method has been applied successfully on reduced-representation datasets before, the sampling density of our target enrichment dataset falls below the threshold for successful analysis. We have included these trials in our methods for readers to find out about our total attempts at species inclusion.

With a greater sample of species to work with, it might then be possible to formally test whether species with the largest predicted shifts in breeding habitat show corresponding changes in estimated Ne. Such a result would substantially bolster the inferential argument presented here about that potential relationship.

We thank the reviewer for the suggestion to quantify and compare the extent of potential breeding areas across the duration of *N_e_* fluctuations provided by stairway plot. We have undertaken substantial additional analysis to perform ecological niche modelling to reconstruct the potential past breeding range of *Numenius* shorebirds (see Materials and methods: Ecological niche modelling). We acquired bioclimatic variables for the present day (1960-1990), mid-Holocene (6,000 years ago) and Last Glacial Maximum (22,000 years ago), with present-day species occurrence data as model input. Model parameters were systematically tested and the model with the highest continuous Boyce index coupled with a low omission rate was selected as the best model. The best model was then projected onto paleo-climatic datasets to reconstruct past breeding distributions for 4 species with sufficient input on present-day breeding occurrence. Total breeding area was then quantified and compared across the three time points (see new Figure 1C). Most importantly, we found that bioclimatic models predict an increase in potential breeding area for the late Pleistocene (~20,000 – 10,000 years ago) when the main decline in population-genetic diversity occurred in most *Numenius* species. This finding considerably strengthens our hypothesis that factors other than historic climate change, especially megafaunal extinction, may have constituted the main driver of *Numenius* shorebird diversity decline.

In terms of presentation, the paper jumps between historical climate change at recent geological time scales, and current anthropogenic climate change. The two extinct curlew species in this clade almost certainly went extinct owing to primary causes other than anthropogenic climate change. Past effects of climate and broad-scale habitat change are certainly relevant, but the chain of causality presented here could be improved.Some statements about past processes are presented as facts, whereas my understanding is that they are still subject to debate. In particular the causality of human hunting>megafaunal extinction>widespread habitat change from grasslands to forest is more of a hypothesis. Similarly, the evidence for lower genetic diversity creating higher extinction risk is fairly nuanced. In presenting their own results, the authors may want to be slightly less declarative about causality, such as in saying "Our study revealed that climatic and environmental changes impacted the genetic diversity of curlews" when the data are corelative and based on only a couple of taxa.In general I recommend focusing more tightly on the inferences about the potential relationship between past Ne and broadscale habitat change during the Pleistocene and thereafter. These are fascinating and intriguing enough on their own, but they are indeed highly inferential. In terms of present and future risk, anthropogenic climate change is probably not among the greatest potential drivers of population declines and possible extinction in this group of shorebirds.

We are grateful for the reviewer’s reminder to improve our presentation of potential chains of causality in our manuscript. We have addressed the reviewer’s concerns in the following ways:

1) We have conducted a thorough review of our manuscript to modify or remove statements that are excessively declarative in their claims. We have revised certain statements (e.g., those suggesting that climatic and environmental change have impacted the genetic diversity of *Numenius*) to be more accurate and less definitive in portraying the findings of our study. We have also removed statements of causative links that are not essential to our discussion (e.g. widespread megafaunal extinction was underway due to … the arrival of early *Homo sapiens* that carried out hunts). We agree with the reviewer that a discussion of the drivers of megafaunal extinction goes beyond the scope of our study, which is more thoroughly focused on the drivers of *Numenius* endangerment, so we have ensured that our manuscript no longer engages in a discussion of such tangential subjects.

2) We have shifted our discussion away from extinction risk and focused it on the observed trends in genetic diversity. Links between genetic diversity loss and extinction risk are supported by an increasing body of literature but – nevertheless – remain controversial, and our discussion of these links is now restricted to a few carefully-worded sentences. In the revised discussion, we have acknowledged that such links are more relevant to small populations and we have provided additional citations.

3) To strengthen our inferences regarding factors that affect demographic history, we have added more climatic, biotic and anthropogenic factors to our hypothesis testing (Figure 1C). For example, in the revised main figure, we can now infer with greater confidence that recent anthropogenic factors do not play a large role in genetic diversity declines as these declines took place before the modern human footprint became pervasive. Of additional significance is that our ecological niche modelling now shows that climatic conditions favoured an increase in breeding distribution of *Numenius* shorebirds after the Last Glacial Maximum at a time when genetic diversity in most species precipitously declined. This is a strong indicator that factors other than natural climate change or direct anthropogenic impacts played a role in genetic diversity declines among *Numenius* species.

Recommendations for the authors:I recommend substantially reworking the abstract to make it less general and to highlight even more the truly novel components of this study.

We have completely rewritten the abstract to focus on our findings surrounding *Numenius* shorebirds. Our abstract highlights the important components of our study, namely that we provide the first complete phylogenomic tree including two presumably extinct species, and that we reveal a generally sharp decline in genetic diversity of *Numenius* shorebirds soon after the Last Glacial Maximum. We have added new ecological niche modelling analyses and a range of biotic and anthropogenic factors that allow us to home in on the potential drivers of genetic diversity declines. In the abstract, we also now discuss with greater clarity that megafaunal extinctions, more so than direct anthropogenic or natural climatic factors, seem to be able to account for genetic diversity declines in *Numenius* shorebirds.

If you decide to add more genomic data, we have two borealis specimens in the Cornell Museum of Vertebrates that I would be happy to sample for you (though I do understand the challenges of adding data, doing the reanalyses, and arranging all of the permits to get samples from here to there…). I really do wish that you could add at least borealis and americanus to the stairway plots, since they each typify one of the different habitats in your general hypothesis.

We are very grateful to the reviewer for the generous offer of precious ancient samples. Despite having sufficient sampling of *Numenius borealis* (n=5) for the application of stairway plot, two samples turned out to be quite degraded and showed high missingness when bioinformatic diagnostics were performed (Materials and methods: Demographic history reconstruction), negatively affecting our inferences when including these samples in stairway plot analysis (new Figure S2). When we ran stairway plots only on samples with low missingness, results were clearly affected by a lack of sufficient sample size (new Figure S2). While we strongly considered adding new samples (including perhaps the ones generously offered by the reviewer), we decided against it in the end due to external constraints that would not allow us to expand the scope of this project in such a fundamental way. We are also cognizant that any historic sample has a chance of being unusable in bioinformatic steps due to DNA degradation. On a positive note, we were able to include the extant species *Numenius americanus* in stairway plot analyses, thereby increasing the representation of North American species and strengthening our comparisons and discussions.

I'm not sure that this is really one of the MOST 'extinction prone' groups of birds, especially in comparison to various island groups.

We have removed the statement as suggested by the reviewer.

Personally I think that the timing of human spread around the globe and megafaunal extinction is too close to be spurious, but that debate goes back decades and I'm not sure there is consensus among the experts on whether it was causal?

We agree with the reviewer that there have been many studies investigating the relative interactions between human spread and megafaunal extinction, with differing conclusions at different geographical scales. As the cause of megafaunal extinction is not crucial to our study and we did not generate data for this question, we have removed this part of the discussion.

Reviewer #2:In this paper, the authors use genomic data to explore how historical processes may have helped shape the current conservation statuses of a group of threatened (and in some cases extinct) migratory birds. They conclude that nearly all of the species in the group have exhibited declines in their effective population size since the Last Glacial Maxima, but that these declines have been particularly large in more temperate breeding species. The authors then attribute the more substantial declines of the temperate breeding species to the larger overlap these species have had with historic anthropogenic activities.This paper has a number of strengths: (1) It is clearly written and easy to follow. (2) It provides the first species-level phylogeny of this group of species. (3) It focuses on a group of species that has already experienced two extinctions and includes a number of other threatened and endangered taxa; as such insights that may contribute to their conservation are sorely needed.

We thank the reviewer for the positive reflections on our study.

The paper also has a number of significant weaknesses. Most importantly: (1) No testable hypotheses or predictions are set forth.

We are grateful for the reviewer’s suggestion to improve the framing of our manuscript. In our revised manuscript, we have made our hypothesis more explicit in the “Introduction”. Our null hypothesis is that diversity declines in *Numenius* shorebirds are largely attributable to direct anthropogenic factors, considering that habitat loss and hunting are major threats that are known to have caused extinction events in this group of migratory birds. At the same time, we also hypothesise that their genetic diversity should closely track the availability of breeding habitat throughout the Late Quaternary. Therefore, we expect to observe that genetic diversity declines would have been most significant during the late Holocene, and would intensify with increasing anthropogenic activity. Coincidentally, our data contradict this null hypothesis and point to other factors being more important. We whole-heartedly agree with the reviewer that a hypothesis framing substantially improves our manuscript.

(2) While sampling across taxon is extensive, sampling within each taxa is more limited and precludes the inclusion of most in the core analyses.

We thank the reviewer for recognising our extensive species sampling of the shorebird genus *Numenius*, allowing us to produce the first complete phylogenomic tree of this genus, including two presumably extinct species. At the same time, we aimed to acquire comprehensive sampling within each taxon through an exhaustive search of suitable samples from museums and collaborators. We recognise that Reviewer #1 had also reflected similar concerns, which we have responded to in detail in Reviewer #1 response #3. In summary, we have now added *N. americanus* to the stairway plot, allowing for the inclusion of 5 out of 9 species. Importantly, the species featured represent extinct and extant species, as well as tundra or temperate-breeding species of both continents, allowing for adequate comparisons.

(3) No effort is made to reconstruct the historic ranges of any of the species or, therefore, to robustly analyze the historical processes that may have been most likely to affect their effective population sizes.

We have taken the reviewer’s suggestion to heart and conducted a substantial volume of additional ecological niche modelling on the species included in the stairway plot analysis for which sufficient occurrence points were available (n=4; Materials and methods: Ecological niche modelling). We acquired bioclimatic variables and species occurrence points in the breeding ranges of *Numenius* shorebirds and input them into Maxent to reconstruct potential breeding distributions. We also quantified the total suitable breeding area at present (1960-1990), mid-Holocene (6,000 years ago) and Last Glacial Maximum (22,000 years ago) (new Figure 1C). With these additional analyses, we were able to directly test which processes may have played a role in impacting the effective population sizes of *Numenius* shorebirds. Effective population sizes generally declined across *Numenius* species in the aftermath of the Last Glacial Maximum, even though this period was characterized by increases in the extent of suitable breeding area, suggesting that climate-induced habitat changes are unlikely to have affected fluctuations in *N_e_*, and hinting at other causes instead.

In the end, I find it difficult to know what conclusions to walk away with. With only five taxa meeting the threshold of five individuals for inclusion in the generation of effective population size 'histories' it's really difficult to run any formal analyses that might allow for robust statistical tests. This then limits the ability of the authors to draw strong conclusions about which groups of species have experienced the strongest declines in genetic diversity.I also urge the authors to read and digest Crisp et al. 2011 in Trends in Ecology and Evolution. That paper provides a framework for how biogeographical studies can generate testable hypotheses, something that this manuscript lacks. Besides 'eyeballing' Figure 1C, it's impossible to assess whether any of the historical processes discussed actually were likely to play a role in the observed declines in effective population size. What alternative hypotheses might there be that could also be tested and refuted?

We appreciate the reviewer’s critical comment and have taken their suggestion on board. The Introduction of our revised manuscript now clearly frames the hypothesis of our study, which is then addressed again in the Discussion. In Crisp et al. (2011), which we have added as a citation, timings of events are used to test hypotheses of unobserved processes. To strengthen our inferences about the factors that may have played a role in *Numenius N_e_* fluctuations, we have adopted the same approach as in Crisp et al. (2011) by including more climatic, biotic, and anthropogenic factors and relating the timing of their occurrence directly to the diversity fluctuations of the stairway plot results (Figure 1C). In brief, the timing of the steep declines of genetic diversity in most *Numenius* species considerably pre-dates the expansion of the modern human footprint on the planet, rendering direct anthropogenic impacts an unlikely culprit of diversity declines. By the same token, natural climate change would have predicted an increase rather than decline in post-LGM genetic diversity. Instead, our reconstructions of *N_e_* fluctuations are consistent with the impact of megafaunal extinctions on habitat maintenance. We have carried out revisions to the entire manuscript to reflect this clear thread of hypothesis testing in response to the reviewer’s concern, and we hope the new manuscript achieves this well.

Reviewer #3:Overall, a study such as the one presented is important for understanding our current biodiversity crisis and I appreciate the work. The methods are thorough and very nicely written.

We thank the reviewer for the positive review of our manuscript.

At this stage based on the framework presented the authors do not satisfactorily present evidence that megafauna decline and climate change drove the reduced population sizes in the focal species. To strengthen the work there need to be testable hypotheses presented and I would recommend moving away from the idea that only the glacial-interglacial end of Pleistocene climate change event is responsible for extinction while ignoring long term human impacts on biodiversity across the entirety of the Holocene (past 11,000 years).The authors use genetic data to evaluate the evolutionary relationships and the population size of species of Numenius (shorebird) species. They found a decline in population sizes starting at the end of the last glacial of the Pleistocene. These declines are comparable to the declines found in megafauna which generally (on continents) occurred at toward the end of the Pleistocene and the end of the last glacial. However, there is not a direct link demonstrated that the declines of these shorebirds were caused by the declines of megafauna (which is what the title states). While many large mammals were ecosystem engineers, and the loss of these species could possibly impact the breeding habitats of these species leading to a reduction of breeding grounds resulting in declines this conjecture there is no data supporting these linkages herein. For example, one could also argue that increased human population size could also be driving these patterns or just the loss of wintering habitat due to higher sea levels and reduced coastline in some areas may be driving these patterns as well. I would argue that you have a more evidence that (as stated in the abstract): "Species breeding in temperate regions, where they widely overlap with human populations, have been most strongly affected". This falls in line with the fact that Eskimo Curlew was decimated by human hunting in the 19th century and was already in decline from the conversion of breeding ground habitat to agricultural land. Human activities since the late Pleistocene (which includes landscape modifications) have led to the current extinction crisis that has occurred since the late Pleistocene. It would be more effective to highlight this more than megafaunal losses driving the population declines in these species. Trophic cascades driving extinction/population reductions are a possibility but without direct evidence it is best left for discussion points.

We acknowledge the reviewer’s important comments, most of which coincide with comments made by previous reviewers (and addressed therein). We reiterate the most important actions that we have taken in response to the reviewer’s points, but we would also like to refer the reviewer to our previous responses to the other two referees (see above). In brief, we have carried out the following revisions in response to the points raised by this reviewer:

1) Firstly, we have defined a clear hypothesis in the “Introduction” and adjusted the entire manuscript around a hypothesis-testing framework (also see Reviewer #2 response #2, Reviewer #2 response #5).

2) Secondly, we have now included more climatic, biotic, and anthropogenic factors throughout the time range of our stairway plot reconstructions. We have also performed additional analyses to quantify the potential breeding area of *Numenius* shorebirds at three points in time to verify if habitat availability correlates with *N_e_* fluctuations (also see Reviewer #1 response #5).

3) Thirdly, we are grateful for the reviewer’s reminder to be mindful when making statements about causality and have reviewed our manuscript to remove broad claims (also see Reviewer #1 response #5, Reviewer #2 response #5).

Based on our enhanced dataset, we are now able to show that declines in *N_e_* considerably pre-date the expansion of the human footprint on the planet, ruling out a major direct impact of recent anthropogenic activities on the *N_e_* decline documented in our study.

Contrary to expectations, anthropogenic factors such as increasing human population size and habitat conversion only rose in significance after the *N_e_* of most *Numenius* species had already declined and stabilised. We cannot rule out that recent anthropogenic activity has had adverse effects on genetic diversity in *Numenius*, but these effects do not yet seem to be captured by our data, possibly on account of shorebirds’ long generation times. These points are now discussed in the revised manuscript version.

To significantly strengthen the argument that climate change is the driving factor of declines I recommend investigating how effective population sizes changed across the Quaternary (past 2.6 million years). The last Pleistocene glacial- modern (Holocene) interglacial climate change event is not a unique and these dynamic conditions have occurred 20+ times across the Quaternary. Therefore, if you find that multiple times across the past 2.6 million years there has been a reduction in effective population size that corresponds to interglacial intervals then perhaps you can state that climate change has led to the decline of Numenius populations. Therefore, only looking at a single time point where there were much more destructive processing taking place i.e., human direct and indirect impacts, I find it is hard to draw this firm conclusion that it is climate change driving these patterns in your data. To better sort out various processes and patterns I would recommend restructuring the introduction to provide the background regarding the history of the Quaternary from a climate, extinction, and human perspective and provide testable hypotheses e.g., 1. Across the past 2.6 million years during interglacial intervals, when, presumably, breeding habitats and shoreline (wintering habitats) were less extensive, effective population sizes were greatly reduced due to limited resources. If this finding is not the case, then you would need another factor to have driven these species to have reduced population sizes at the end of the Pleistocene-Holocene and throughout the Holocene.

We understand the reviewer’s suggestion to compare *N_e_* fluctuations across multiple cycles of Quaternary cooling and warming to conclusively pin down the role of climate change in affecting genetic diversity. However, we are afraid that testing for such cyclical changes in genetic diversity is beyond what we would be able to achieve with our dataset of genome-wide loci, especially considering the computational complexity in accounting for multiple glacial cycles. Additionally, there is also very scant bioclimatic data for cycles that pre-date the LGM.

We echo the reviewer’s sentiment that natural climate change should not be credited too readily with having exerted such a strong impact on genetic diversity in shorebirds, and we would like to point out that our manuscript’s main conclusions actually agree with this notion. After adding novel analyses using Maxent to model the extent of suitable breeding area of multiple species at present (1960-1990), mid-Holocene (6,000 years ago) and Last Glacial Maximum (22,000 years ago), we found that bioclimatic data predict a post-LGM increase in breeding area for most species (even though these same species experienced a decline in genetic diversity). Therefore, we are in agreement with the reviewer that natural climate change is unlikely to have resulted in the genetic decline, and that other factors must be responsible. We hope the revised version of the manuscript satisfactorily addresses these reviewer criticisms.

Overall, human impacts especially highlighting what happened to the Eskimo Curlew as hunting (loss of millions of individuals), breeding habitat destruction for agriculture, and the extinction of its main food item on the breeding ground (Rocky Mountain locust) should be included to strengthen the work. You can even provide human population size estimates, as available, in various parts of the breeding ranges across the Holocene.

As suggested by the reviewer, we have now added more anthropogenic factors and provide this data in greater detail (new Figure 1C). We have now incorporated human population size estimates for North America and Eurasia from HYDE 3.2. We have also added the corresponding agricultural land use estimates from HYDE 3.2. Both measures of anthropogenic impact showed great increases only in the most recent millennium, while genetic diversity declines substantially pre-date this period of human impact. We are able to infer with confidence that the main factors causing the steep genetic diversity declines documented in our study are unlikely to be due to direct anthropogenic impacts, although such impacts may have exacerbated the situation for temperate-breeding species (see new Discussion).

I appreciate this project and manuscript and the acknowledgement that we need to understand declines and extinction of species considering the current diversity crisis.

We thank the reviewer for affirming the value of our manuscript.

Recommendations for the authors:1. Title is too general for the study and the results.

We have amended the title to focus on genetic diversity declines in *Numenius* shorebirds.

2. The climate change event that occurred at the end of the Pleistocene and beginning of the Holocene was not a unique event. These glacial-interglacial intervals occurred 20+ times across the past 2.6 million years. Accordingly, it is important to evaluate whether there were multiple reductions of effective population size in these species across this time. This would strengthen the argument that climate change is impactful on these shorebirds' distributions and population size. If not then another factors need to be evaluated.

We agree with the reviewer that climate oscillations have occurred many times throughout the Pleistocene. The reviewer had raised this point in the introductory remarks, and we have addressed it there (see Reviewer #3 response #3). In summary:

1) While we agree that a demonstration of correlation across multiple glacial cycles would be valuable, we are not certain that our dataset (and many other datasets available) would allow for such complex demonstrations.

2) Our data refute the notion that natural climate change has impacted *Numenius* shorebird trajectories. Given that this correlation could not be established even for the time after the Last Glacial Maximum, perhaps showing such a correlation for previous glacial maxima becomes moot.

Line 107 – Is the decline seen in differential hunting of various species after 700 years ago?

We have considered the reviewer’s suggestion that differential hunting pressures may play a role in the differential declines of curlews versus whimbrels in recent years. However, we will not be including this discussion point as there is too little information regarding differential hunting pressures (Pearce-Higgins et al., 2017) and cooccurring *Numenius* species may be easily confused with one another by hunters (Jiguet et al., 2021). Moreover, we have since removed our discussion of their differential declines to be more conservative in our inferences.

Figure 1 caption – It looks like the demographic modeling wasn't performed for more than half of your focal species. This should be explicitly stated in the methods/results/discussion for transparency.

With the inclusion of *N. americanus*, we now have more than half of our focal species represented in our demographic modeling. Nonetheless, we thank the reviewer for the reminder to state this more explicitly in our manuscript. We have added a detailed discussion of our criteria for a species’ inclusion in our demographic modeling (see Materials and methods: Demographic history reconstruction).

Line 150 in the figure caption there is a missing space

We have amended the figure caption to include the necessary space.

For the historical specimen DNA extraction methods please briefly elaborate and provide details on how you modified the Qiagen kit for the historical specimen extractions. Currently there is a citation but please briefly include some extra details so folks can easily access them.

We have included descriptions of the modifications made to the Qiagen kit for DNA extraction of historic samples (Materials and methods: Laboratory methods).

Line 263 – is the perl script for removing potential linked SNPs available somewhere or upon request? Might be worthwhile mentioning that it is to strengthen the methods.

We have included the name of the perl script, alongside the citation and link to its github repository (Materials and methods: SNP calling).

[Editors’ note: what follows is the authors’ response to the second round of review.]

Essential revisions:As the reviewing editor, I am supportive of the publication of this paper in its current form. However, Reviewer 1 still has reservations about the power of inference here for inferring a causal effect of the megafaunal extinctions. The way eLife works, we have had a dialogue about these disconnects, and even after that back-and-forth we are in different places about the inferential power of this study. To reconcile these different levels of comfort with that inference, please consider the following small changes:1) in the title, change "seem to" to "may"

We have made the suggested change to the title of our manuscript.

2) add an additional qualifier to the abstract to indicate that the megafaunal scenario is one of multiple potential explanations.

In our abstract, we have added the phrase “…among other factors…” before our discussion of the potential impact of megafaunal extinctions, and made additional smaller wording changes to indicate that there are other factors that can explain genetic diversity loss in *Numenius* shorebirds (Abstract, line 37).

Reviewer #1 (Recommendations for the authors):I commend the authors on their revised effort and greatly appreciate the addition of the ecological niche modeling, as well as explicit hypotheses to be tested. To my eyes the modeling appears robust. The paper is also clearly written and flows nicely from idea to idea.Where I continue to struggle, however, is with the inferential power of the study. I apologize, but I cannot quite move past the fact that, with only five species, it is impossible to run any statistical analyses linking changes in Ne with changes in habitat availability or any other historical factor.

We thank the reviewer for their thoughtful feedback and we acknowledge that the sample size of our study precludes statistical analyses that may increase our inferential power. We have screened our manuscript to remove any remaining instances where we may have used strong language to build a causal connection between megafaunal extinctions and shorebird diversity loss (for example, Results and discussion, line 164 “…may have had cascading effects …”). Our manuscript now consistently employs cautious wording in drawing on any such potential associations.

– For instance, Reviewer #3 brought up in the previous round of comments the possibility that rising sea levels reduced stopover/nonbreeding habitat simultaneous with the megafaunal extinction. That possibility is not addressed with any analyses, and quite possibly can't be addressed given the limited sample sizes involved, but it should nonetheless be mentioned.

We have taken the reviewer’s suggestion on board and added a sentence discussing that the impact of rising sea levels on the availability of non-breeding habitats for *Numenius* remains to be investigated (Results and discussion, lines 165–169). However, we have also added our perspective that rising sea levels after the LGM changing non-breeding habitat availability are unlikely to have explained post-LGM declines in genetic diversity in *Numenius* shorebirds. Sea level rises have increased the total length of shorelines, at least in the Old World, but quite possibly also across the planet, through the generation of complex archipelagos (e.g. Indonesia) via sea water immersion in shelf areas (De Groeve et al., 2022; Sarr et al. 2019). Therefore, rising sea levels are likely to have increased rather than reduced the availability of non-breeding habitat and we consider it unlikely that its incorporation would have changed any of our conclusions in this study. Nonetheless, our discussion now briefly touches upon this subject and calls for this factor to be taken into account in future studies.

– What is more, the apparent disconnect between the ecological niche modeling results and Ne results, gives me pause. While the reason offered is plausible, it's hard not to get stuck there without some sort of actual analysis to help explain the difference.

We have expanded our discussion to further address the discrepancy between our ecological niche modelling and *N_e_* results (Results and discussion, lines 115–121). While we initially shared the reviewer’s consternation, we do believe that this disconnect forms the very heart piece of our conclusions, indicating that climate change alone does *not* explain *Numenius* diversity declines, and instead offering alternative explanations such as megafaunal extinctions. Our expanded discussion now also offers an alternative explanation that at least part of the discrepancy could be explained by rapid range expansion, as predicted by an increase in suitable habitat between LGM and midHolocene, resulting in decreases in *N_e_*. However, *N_e_* continued to decrease when predicted suitable habitat remained relatively constant after the mid-Holocene, suggesting that other factors (such as megafaunal extinction) would need to be considered to understand this discrepancy.

– Finally, I am struck by the dramatic differences in Ne patterns between N. phaeopus and N. hudsonicus given their biological and ecological similarity. Is this an artifact of only having 5 N. hudsonicus? (I could provide you with dozens of samples if you would like!) In either case, some mention of this discrepancy in patterns should at least be mentioned in the main text.

We are thankful to the reviewer for specifically pointing out the discrepancy in demographic history between the two sister species *N. phaeopus* and *N. hudsonicus*, which is indeed pronounced and requires an explanation. Our previous manuscript version did not offer any explanation as we feared this may be seen as too taxon-specific, but we have been happy to add a likely explanation of this pattern in the present draft version (Results and discussion, lines 150–154). We also thank the reviewer for generously offering us additional samples for *N. hudsonicus*, but we do not believe that additional samples are necessary this late in the process, as our sample sizes should be sufficient for the conclusions we drew. Rather than attributing these differences to sample size, we believe the discrepancy in pattern between the two whimbrels can easily be explained by the differences in glacial extent and impact between North America and Asia. North America was covered by vast ice sheets during the LGM while most of Siberia remained ice-free, allowing for disproportionately high levels of N(e) in *N. phaeopus* in comparison with *N. hudsonicus*.